# Progressive enhancement of kinetic proofreading in T cell antigen discrimination from receptor activation to DAG generation

**Derek M Britain, Jason P Town, Orion David Weiner***

Cardiovascular Research Institute and Department of Biochemistry and Biophysics, University of California, San Francisco, San Francisco, United States

**Abstract** T cells use kinetic proofreading to discriminate antigens by converting small changes in antigen-binding lifetime into large differences in cell activation, but where in the signaling cascade this computation is performed is unknown. Previously, we developed a light-gated immune receptor to probe the role of ligand kinetics in T cell antigen signaling. We found significant kinetic proofreading at the level of the signaling lipid diacylglycerol (DAG) but lacked the ability to determine where the multiple signaling steps required for kinetic discrimination originate in the upstream signaling cascade (Tiseher and Weiner, 2019). Here, we uncover where kinetic proofreading is executed by adapting our optogenetic system for robust activation of early signaling events. We find the strength of kinetic proofreading progressively increases from Zap70 recruitment to LAT clustering to downstream DAG generation. Leveraging the ability of our system to rapidly disengage ligand binding, we also measure slower reset rates for downstream signaling events. These data suggest a distributed kinetic proofreading mechanism, with proofreading steps both at the receptor and at slower resetting downstream signaling complexes that could help balance antigen sensitivity and discrimination.

**\*For correspondence:**
orion.weiner@ucsf.edu

**Competing interest:** The authors declare that no competing interests exist.

## Editor's evaluation

The authors have used an ontogenetic approach to advance understanding of the kinetic proof-reading steps down-street of the antigen receptors. The authors have provided evidence for longer lifetimes of down-stream signaling events in relation to physiological receptor-ligand binding kinetics. This accounts for the ability of immune cells to respond across a large range of agonist pMHC while retaining the ability to prevent responses to very short lived interactions associated with self ligands. The results advance our knowledge of chimeric antigen receptor signaling and provides an approach to identify optimal receptor binding kinetics to couple to down-stream signaling machinery.

## Introduction

Proper antigen discrimination is a cornerstone of the adaptive immune system. Failure of T cells to activate in response to foreign antigen can enable pathogens to invade the body undetected (*Charles A Janeway et al., 2001*; *Horst et al., 2011*). Conversely, improper T cell activation against self-antigen causes autoimmune disorders (*Dornmair et al., 2003*). When a T cell contacts an antigen-presenting cell (APC), it can detect the presence of 1–10 molecules of foreign antigen (*Christinck et al., 1991*; *Demotz et al., 1990*; *Kimachi et al., 1997*; *Sykulev et al., 1996*), despite the much

**Figure 1.** Kinetic proofreading discriminates binding events by half-life. (**A**) The kinetic proofreading model consists of a chain of slow sequential events ($C_i$ with rates $k_i$) that begin once the ligand binds the receptor ($k_{on}$). The chain of events must be completed before a productive signal is communicated to the cell. When the ligand unbinds the receptor ($k_{off}$), all accumulated signaling intermediate ($C_i$) reset back to the ground state. Only binding events that persist long enough to form the active terminal signaling complex ($C_{final}$) result in cell activation. (**B**) Long-lived binding events remain bound to the receptor for a sufficient duration to complete all kinetic proofreading steps, resulting in a productive signal. (**C**) Short-lived binding events unbind the receptor before the kinetic proofreading chain is completed and fail to fully stimulate the antigen signaling cascade.

greater abundance (×100,000 or more) of self-antigen (*Bhardwaj et al., 1993*; *Cohen et al., 2003*; *Irvine et al., 2002*; *Unternaehrer et al., 2007*). These observations indicate that T cells must use parameters other than the number of ligand-bound receptors to distinguish self from foreign antigen (*Daniels et al., 2006*; *Davis et al., 1998*; *Gascoigne et al., 2001*; *Germain and Stefanová, 1999*).

One attractive model for how T cells discriminate antigen is the kinetic proofreading model (*Hopfield, 1974*; *McKeithan, 1995*; *Ninio, 1975*), where only antigens that continuously bind a TCR for a sufficient duration activate the T cell (*Figure 1A*). Kinetic proofreading postulates that antigen

binding initiates a series of sequential biochemical events that are substantially slower than antigen-binding lifetimes and must progress to completion before activating the T cell. Only antigen that stays bound to the TCR long enough for the completion of all events is stimulatory to the T cell (*Figure 1B*). If the antigen unbinds the TCR before all events are complete, the system resets back to the ground state (*Figure 1C*). While it is commonly accepted that T cells use kinetic proofreading for antigen discrimination (*Coombs and Goldstein, 2005*; *Courtney et al., 2018*; *McKeithan, 1995*), we do not know which steps in the antigen signal transduction cascade enable the strong kinetic proofreading observed in T cell activation.

Kinetic proofreading can convert small differences in ligand-binding lifetime into large differences in cellular output. The number of proofreading steps sets the strength of a proofreading system, with stronger proofreading systems requiring a higher number of proofreading steps. Ten steps of kinetic proofreading would roughly amplify a twofold difference in ligand-binding lifetime into a 1000-fold difference in output. Kinetic proofreading allows T cells to ignore numerous shorter-lived self-antigen-binding events, while activating in response to rare longer-lived foreign antigen-binding events (*Davis et al., 1998*; *Gascoigne et al., 2001*; *Germain and Stefanová, 1999*).

Previously, our group developed a light-gated immune receptor that we used to probe the role of ligand kinetics in T cell antigen signal transduction (*Figure 2A*). We found evidence of proof-reading at the level of DAG generation, while detecting no proofreading at the recruitment of Zap70 to the activated receptor (*Tischer and Weiner, 2019*). The strength of proofreading measured at DAG generation indicated multiple upstream proofreading events. Our optogenetic system requires a light-insensitive mode of cell adhesion to ensure reliable measurement of signaling activity readouts. However, our previous adhesion method was not compatible with robust cellular activity readouts upstream of DAG generation, preventing us from identifying where the multiple upstream proof-reading events originate.

Here, we improve our assay by incorporating the native integrin ICAM-1 as our light-insensitive mode of adhesion, which improves the robustness of our previous biosensors and enables the use of new biosensors (*Bromley and Dustin, 2002*). With integrin adhesion, we now detect kinetic proof-reading in receptor activation upstream of the recruitment of Zap70. We measure stronger upstream kinetic proofreading at the clustering of the scaffold protein LAT, suggesting proofreading steps between Zap70 recruitment and LAT cluster formation. Furthermore, we measure even stronger kinetic proofreading at DAG generation, suggesting additional proofreading steps after the initial formation of LAT clusters. We also find that LAT clusters reset slower than Zap70 recruitment upon unbinding antigen. Our results suggest a kinetic proofreading system that starts at the level of receptor activation and continues across multiple spatially segregated signaling complexes, with downstream signaling complexes resetting slower than receptor-level signaling complexes. Such a system would allow high concentration of antigens with intermediate binding lifetimes to activate T cells, while still responding to rare long binding antigens and filtering out short binding antigens, which is consistent with TCR cooperativity postulated in tissue homeostasis and autoimmunity (*Cameron et al., 2013*; *Goyette et al., 2022*; *Korem Kohanim et al., 2020*; *Lin et al., 2019*; *Pettmann et al., 2021*; *Wang et al., 2020*).

## Results

### Expanding suite of optogenetically controlled T cell biosensors by optimizing cell adhesion system

Previously, we developed a light-gated immune receptor (zdk-CAR) to probe T cell antigen signaling output as a function of both receptor occupancy and average ligand-binding half-life (*Figure 2A*; *Tischer and Weiner, 2019*). In the dark, cells expressing zdk-CAR bind to purified LOV2 ligands functionalized on a supported lipid bilayer (SLB) and activate antigen signal transduction. Blue light interrupts LOV2-binding events and antigen signaling. If we provide LOV2 as the only means of cell interaction with the bilayer, cells unbind from the bilayer upon exposure to blue light (*Figure 2B*). This complicates our analysis, which requires exposing cells to multiple doses of blue light to build up a dose-response curve. Therefore, we must also include a light-insensitive mode of adhering the cells to the bilayer.

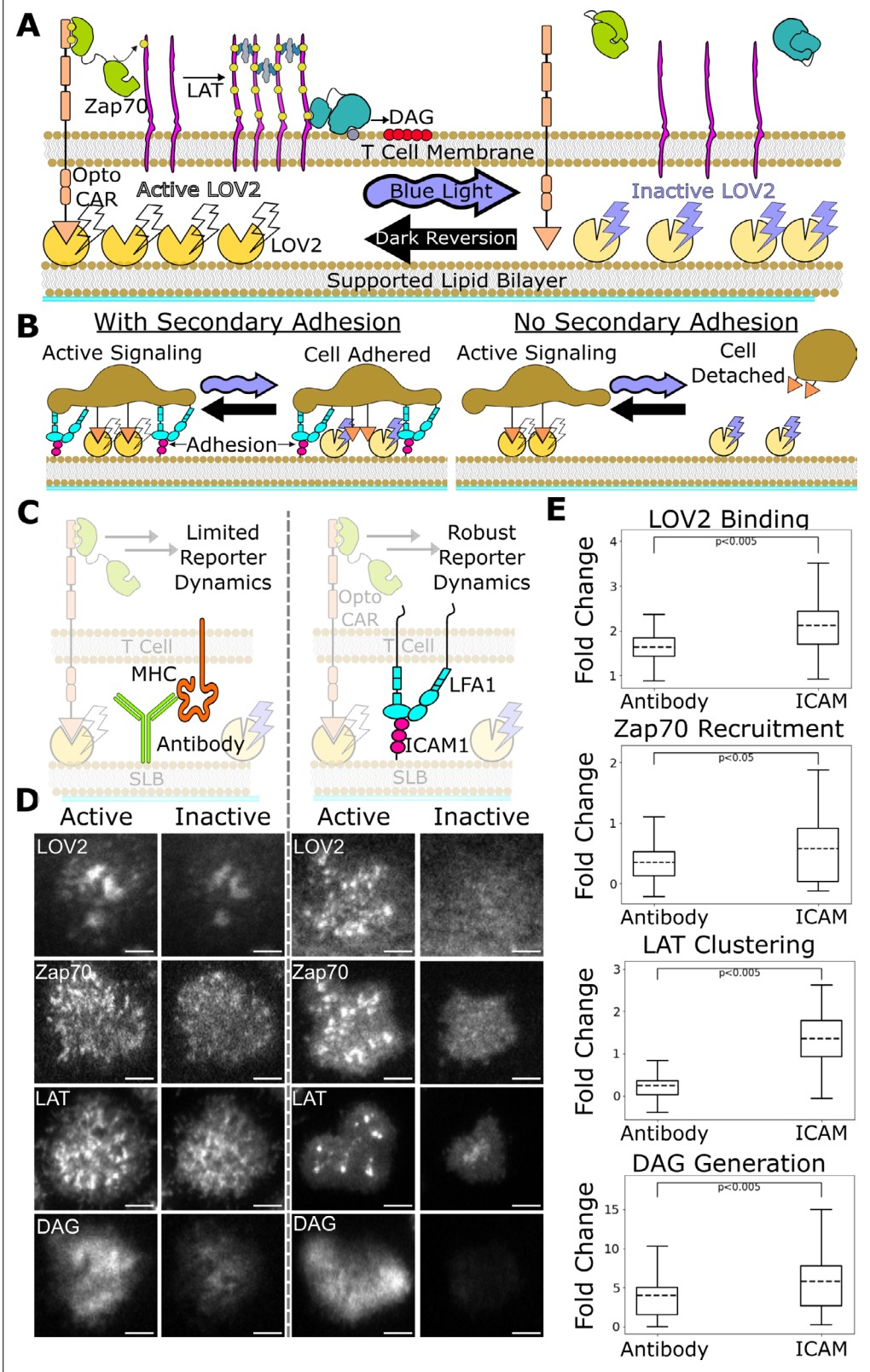

**Figure 2.** ICAM-1 adhesion improves robustness of proximal signaling live-cell biosensors. (**A**) With our optogenetic system, we can control the concentration and lifetime of ligand-receptor interactions with blue light. A supported lipid bilayer (SLB) functionalized with the light-sensitive protein LOV2 creates an artificial antigen-presenting surface. Jurkat cells expressing zdk-CAR bind dark-state LOV2 and activate antigen signaling. Blue light

*Figure 2 continued on next page*

*Figure 2 continued*

interrupts LOV2-CAR-binding events and terminates signaling. (**B**) Our assay requires a secondary light-insensitive mode of cell adhesion to maintain segmentable and trackable cell footprints throughout an experiment. Cells adhere to the SLB through LOV2 while actively signaling. However, when signaling is inactivated by blue light, cells detach from the bilayer if no secondary method of constant adhesion exists. (**C**) In our previous work, we adhered cells to the SLB through biotinylated beta2 microglobulin antibodies against the cells' MHC class1 (left). This adhesion technique limited the spatiotemporal dynamics of many biosensors, including the recruitment of ZAP70 and the clustering of LAT. We now adhere cells to the bilayer through native LFA-1 to ICAM-1 integrin signaling (right). (**D**) Representative TIRF images of LOV2 binding, Zap70 recruitment, LAT clustering, and DAG generation during ligand engagement and after 2 min of strong inactivation of the Zdk-CAR under antibody adhesion (left) versus ICAM-1 adhesion (right) (scale bar = 5µm). (**E**) SLB adhesion through ICAM-1 improved the spatiotemporal dynamics of each biosensor. Boxplot of the maximum fold-change in LOV2 binding, Zap70 recruitment, LAT clustering, and DAG generation between active and inactive Zdk-CAR signaling states under antibody adhesion and ICAM-1 adhesion. Dashed line indicates population mean, each plot represents reporter change of approximately n=120cells, independent t-test p-values shown.

In our previous work, we adhered cells to the SLB using an antibody against their own MHC (Abcam ab21899) (*Figure 2C*, left). However, this method was not compatible with sensitive detection of some cellular activity readouts, particularly the most proximal ones like Zap70 recruitment. In this work we improve our assay by adhering the cells to the SLB through native ICAM-1 integrin adhesion, which generates large cellular footprints while being compatible with a larger suite of antigen signaling biosensors.

We functionalized our SLB with human ICAM-1 to better mimic the T cell/APC interaction (*Figure 2C*, right; *Dustin et al., 2007*). We first validated Zap70 recruitment, LAT clustering, and DAG generation biosensors expressed in Jurkat cells on ICAM-1 functionalized bilayers (*Figure 2D*). Jurkat cells stably expressing zdk-CAR were exposed to SLBs functionalized with purified human ICAM-1-His and Alexa Fluor 488 labeled LOV2. To increase integrin binding, we changed our imaging buffer to a modified HBSS (mHBSS) buffer with higher $Mg^{++}$ concentrations (*Labadia et al., 1998*). Compared to our previous antibody-based adhesion, Jurkats on ICAM-1 functionalized bilayers released a greater percentage of the bound LOV2 ligand upon blue-light illumination. Adhesion through ICAM-1 also increased the dynamic range of the Zap70, LAT, and DAG biosensors (*Figure 2E*; *Videos 1–4*). ICAM-1 adhered cells also displayed receptor/Zap70 and LAT clustering spatial patterns similar to those observed by others using native receptor ligands on SLBs (*Balagopalan et al., 2015*; *Chakraborty and Weiss, 2014*; *Kumari et al., 2015*).

## Quantifying kinetic proofreading in Zap70 recruitment, LAT clustering, and DAG generation under ICAM-1 adhesion

With our improved ICAM-1 adhesion method and expanded suite of biosensors, we applied our previous assay for measuring the strength of upstream kinetic proofreading to the signaling events of Zap70 recruitment, LAT clustering, and DAG generation. In our assay, we independently probe the effect of receptor occupancy or ligand-binding half-life on a given T cell intracellular signal (*Tischer and Weiner, 2019*). By titrating the intensity of LOV2 stimulating light, we modulate the lifetime of receptor-LOV2-binding events and the downstream antigen signaling response. By repeating the experiment at different levels of LOV2 ligand on the bilayer, we can uncouple the effects of ligand occupancy from ligand-binding half-life on cell activation.

After exposing zdk-CAR expressing cells to a LOV2 and ICAM-1 functionalized SLB, we illuminate the cells with a constant intensity of blue light with a known ligand-binding half-life (*Figure 3A*). After 3 min of illumination (when a steady state is reached), we image the amount of bound fluorescently tagged LOV2 accumulated underneath each cell using a long exposure image (*O'Donoghue et al., 2013*). We also image the output intensity of the signaling biosensor. We then reset the system with 2 min pulse of intense blue light to terminate all signaling before proceeding to expose the cells to the next intensity of blue light. We repeat the experiment on bilayers functionalized with different densities of LOV2 to sample a wider range of occupancies. Following this experimental protocol, we build up a dataset of steady-state signaling output as a function of both ligand occupancy and ligand half-life (*Figure 3B*).

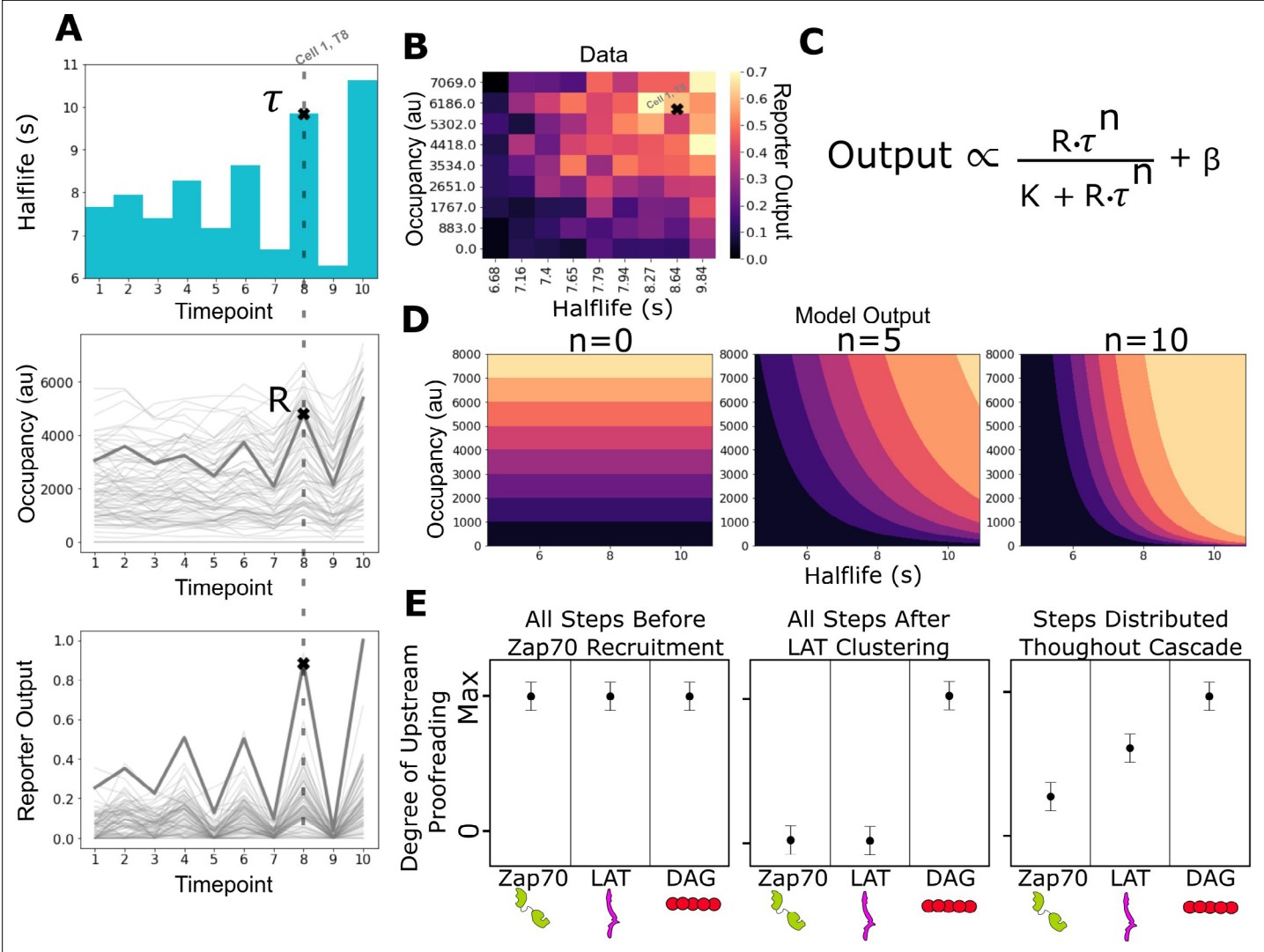

**Figure 3.** Measurement of receptor occupancy, binding half-life, and signaling output for evaluating proofreading strength. (**A**) After adhering the cells to the functionalized supported lipid bilayers (SLBs), we expose them to a series of 3 min blue-light illuminations generating defined average ligand-binding half-lives. At the end of each half-life condition, we measure every cell's steady-state reporter output and receptor occupancy (example cell highlighted at timepoint 8). Following this protocol, we measure every cell's reporter output (bottom) as a function of receptor occupancy (middle) and average ligand-binding half-life (top). (**B**) After normalizing each cell to its average basal reporter activity, the data from multiple time courses with different ligand densities are aggregated, and the reporter output values are normalized to the 90th percentile cell in the dataset. (**C**) The dataset is then fit to a model of the expected output of a kinetic proofreading signaling system. In the model, the expected signaling output is proportional to the ligand occupancy (**R**) and ligand-binding half-life ($\tau$) raised to the number of strong proofreading steps (**n**) upstream of that signaling step. The magnitude of n provides a relative measure of the strength of kinetic proofreading between ligand binding and a given signaling step. K is the amount of upstream signal for half-maximal response, and β is basal signaling in the pathway (**D**) Expected output of models with zero, five, and ten kinetic proofreading steps. As the value of **n** increases, the dependence of output on ligand half-life increases, while the dependence on receptor occupancy decreases. A system with no proofreading responds only to receptor occupancy, while a system with a high degree of proofreading (n=10) is insensitive to numerous short binding events while fully responding to very few long binding events. (**E**) Anticipated results for different scenarios of proofreading step distribution.

The online version of this article includes the following figure supplement(s) for figure 3:

**Figure supplement 1.** Simulations of ligand discrimination with and without kinetic proofreading reset pathways.

The lentiviral vectors we used to express our zdk-CAR and live-cell biosensors result in cell-to-cell variability in the expression level of each construct, even after sorting for cells with CAR expression levels similar to native TCR expression in T cells (*Tischer and Weiner, 2019*). To account for variability in biosensor expression, we normalize each cell to its baseline biosensor intensity measured in the

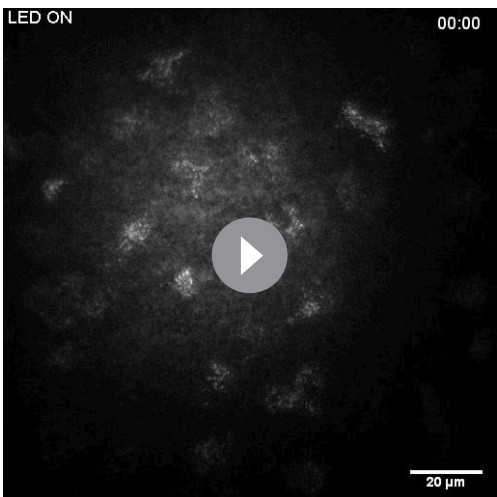

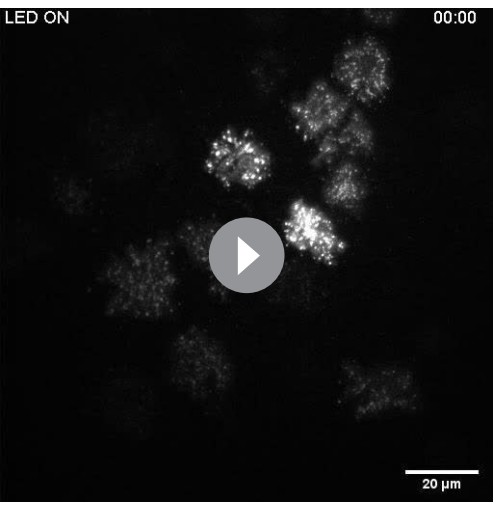

**Video 1.** ICAM-based adhesion yields reversible LOV2 binding for optogenetically stimulated cells. Time course of LOV2 on a supported lipid bilayer reversibly binding zdk-CARs expressed in Jurkat cells in the presence of ICAM-1 adhesion to the bilayer (light-independent cellular interaction). LOV2 is released from cell footprint under blue light and rebinds in the dark (illumination status indicated in top left). Cells were adhered to the bilayer for 5 min prior to the start of the video. Two minutes of strong blue-light illumination are followed by 3 min in the dark for two cycles (MM:SS time-stamp in top right).

https://elifesciences.org/articles/75263/figures#video1

**Video 2.** ICAM-based adhesion yields reversible Zap70 recruitment for optogenetically stimulated cells. Time course of reversible Zap70 recruitment in Jurkat cells in the presence of ICAM-1 adhesion to the bilayer (light-independent cellular interaction). ZAP70 is recruited to phosphorylated zdk-CAR ITAMs in the dark and released back to the cell cytosol under blue light (illumination status indicated in top left). Cells were adhered to the bilayer for 5 min prior to the start of the video. Two minutes of strong blue-light illumination are followed by 3 min in the dark for two cycles (MM:SS time-stamp in top right).

https://elifesciences.org/articles/75263/figures#video2

absence of ligand engagement. The cell-to-cell variability in zdk-CAR expression has the benefit of expanding the receptor occupancies that we sample, as cells expressing more zdk-CAR receptors bind more LOV2 under all half-life conditions (*Figure 3A*, middle). If all cells expressed identical amounts of receptor, it would be laborious to generate and analyze the large number of independent CAR expressing clones needed to sample a wide range of receptor occupancies. Importantly, the variability in receptor expression is not unique to our CAR system; native T cells also vary in TCR expression level and can respond to antigen even at very low levels of TCR expression (*Labrecque et al., 2001*). Kinetic proofreading could enable cells to tolerate variability in receptor expression level, as it results in cells that are increasingly sensitive to ligand-binding half-life instead of receptor occupancy.

We are unable to control when binding events start since our optogenetic system is inhibited by blue light, as opposed to being activated by blue light. The initiation of binding after blue-light inhibition is a function of both the stochastic relaxation of inhibited LOV2 back into the binding state and the diffusion of binding-state LOV2 from outside the previously illuminated area. Without temporal control over the start of binding, it is difficult to measure the time delay between ligand binding and a downstream signaling event (*Yi et al., 2019*). Such studies typically require careful single-molecule imaging of numerous stochastic binding events (*Lin et al., 2019*).

To overcome this technical limitation of our system, we chose instead to measure the steady-state output of the antigen signaling cascade achieved several minutes after ligand binding. Kinetic proofreading systems behave differently than non-proofreading systems at steady state. A non-proofreading system's steady-state output is set by the number of ligand-bound receptors and not the binding half-lives of those ligands (*Figure 3D*, left; *Figure 3—figure supplement 1A*). In contrast, a kinetic proofreading system can produce different steady-state outputs in response to ligands of different binding half-lives, even when ligand densities are adjusted to achieve equivalent occupancy (*Daniels et al., 2006*; *Figure 3D*, right; *Figure 3—figure supplement 1B*). Signaling events take varying amounts of time to occur after ligand binding (*Lin et al., 2019*; *Yi et al., 2019*). However, the

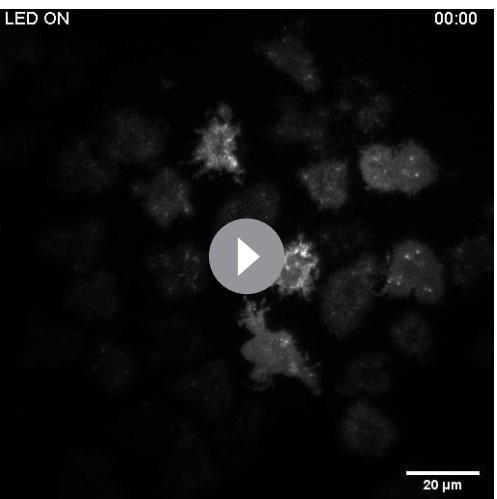

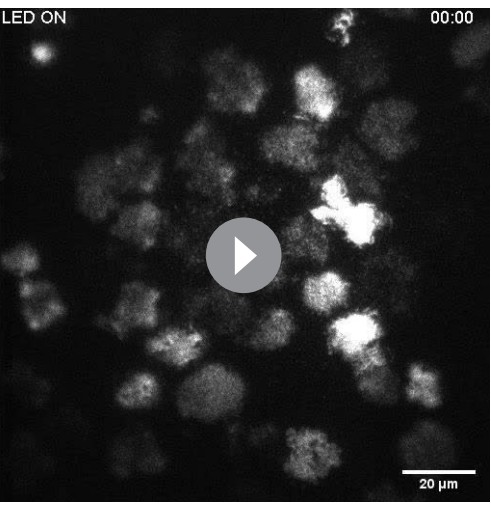

**Video 3.** ICAM-based adhesion yields reversible LAT clustering for optogenetically stimulated cells. Time course of reversible LAT clustering in Jurkat cells in the presence of ICAM-1 adhesion to the bilayer (light-independent cellular interaction). LAT forms clusters during active antigen signaling in the dark, and these clusters dissociate following blue-light-mediated dissociation of the receptor-ligand interaction (illumination status indicated in top left). Cells adhered to the bilayer for 5 min prior to the start of the video. Two minutes of strong blue-light illumination are followed by 3 min in the dark for two cycles (MM:SS time-stamp in top right).

https://elifesciences.org/articles/75263/figures#video3

**Video 4.** ICAM-based adhesion yields reversible diacylglycerol (DAG) generation for optogenetically stimulated cells. Time course of reversible DAG generation in Jurkat cells in the presence of ICAM-1 adhesion to the bilayer (light-independent cellular interaction). DAG is generated during active antigen signaling in the dark, and DAG generation is terminated following blue-light-mediated dissociation of the receptor-ligand interaction (illumination status indicated in top left). Cells adhered to the bilayer for 5 min prior to the start of the video. Two minutes of strong blue-light illumination are followed by 3 min in the dark for two cycles (MM:SS time-stamp in top right).

https://elifesciences.org/articles/75263/figures#video4

temporal delays between steps are on the order of tens of seconds. By imaging the cells after minutes of constant exposure to a set ligand-binding half-life, we measure the steady-state output achieved at a signaling event in the cascade on a longer timescale than these delays (*Tischer and Weiner, 2019*).

Next, we fit our data to a simple model of the expected output from a kinetic proofreading signaling system at steady state (*Figure 3C*). In the model, signaling output is a function of receptor occupancy (R) multiplied by ligand-binding half-life ($\tau$) raised to the number of strong proofreading steps (n). The fit number of proofreading steps indicates how much a reporter's steady-state output depends on binding half-life compared to receptor occupancy. In a system with no proofreading steps, the output only depends on receptor occupancy, and binding half-life becomes irrelevant (*Figure 3D*, left). A system's output with a moderate number of proofreading steps (e.g. five steps) depends on both receptor occupancy and the half-life of ligand binding (*Figure 3D*, middle). A large number of proofreading steps (e.g. 10 steps) results in a system whose output is dominated by ligand-binding half-life and is relatively insensitive to receptor occupancy (*Figure 3D*, right).

Our model represents the expected output of a kinetic proofreading signaling network at steady state. It assumes a network of sequential and non-reversible biochemical events where the rate of progression to the next step is much slower than the rate of reset upon ligand unbinding. We cannot determine the exact number of proofreading steps in the network, as the output of many weak proofreading steps could behave the same as the output of a few strong proofreading steps. However, if we assume equivalent strength for all proofreading steps, we can estimate a lower limit of the number of steps. Our model captures the expected output of a saturable signaling step downstream of such a proofreading system. When we apply our model to a measured signaling event in our system, we assume that event is downstream of the proofreading cascade and ask how many upstream steps of equivalent proofreading best explain its output.

With the assumptions of the model, the fit value of n is unlikely to represent the absolute number of upstream biochemical proofreading steps. However, we can compare the measured values of n between signaling events to understand how the strength of proofreading changes as we progress through the cascade. A higher measured value of n at an event suggests a stronger degree of upstream kinetic proofreading. If we subtract the fit n value of a downstream event from an upstream event, we get the measured increase in proofreading strength between the two events. For example, if we have the network $C_0 C_x$ and measure n=3 at $C_0$ and n=5 at $C_x$, subtracting $C_0$ from $C_x$ results in a proofreading strength increase of 2, indicating additional kinetic proofreading between $C_0$ and $C_x$.

Comparing the measured degree of proofreading at multiple known signaling events allows us to bracket where kinetic proofreading steps exist (*Figure 3E*). If all proofreading steps occurred between ligand binding and Zap70 recruitment, we would measure the same degree of upstream proofreading at Zap70 recruitment, LAT clustering, and DAG generation (*Figure 3E*, left), as there would be no additional increase in proofreading strength beyond Zap70. If all proofreading steps occurred after LAT clustering and before DAG generation, we would measure no degree of upstream proofreading at Zap70 recruitment or at LAT clustering, and see a large jump in the degree of upstream proofreading at DAG generation (*Figure 3E*, middle). If proofreading steps were distributed from before Zap70 recruitment to after LAT clustering, we would measure a non-zero degree of upstream proofreading at Zap70 recruitment with an increasing degree of upstream proofreading as we progress down the cascade to LAT clustering and DAG generation (*Figure 3E*, left). As we will show, this last model is most consistent with our data.

## Kinetic proofreading steps exist between ligand-receptor binding and Zap70 recruitment

In our previous work, we failed to detect kinetic proofreading in Zap70 recruitment (*Tischer and Weiner, 2019*). Using ICAM-1 functionalized bilayers, we now measure moderate amounts of kinetic proofreading in the recruitment of Zap70 to activated receptors. Zap70 recruitment correlated with both receptor occupancy (ρ=0.42) and ligand-binding half-life (ρ=0.49) (*Figure 4—figure supplement 1A*). Our model indicates a moderate degree of proofreading at Zap70 recruitment across three datasets (n=4.5 ± 0.4) (*Figure 4A*). These data suggest the existence of kinetic proofreading steps between ligand binding and Zap70 recruitment.

Previously, we measured Zap70 with no light-insensitive adhesion to the bilayer, as antibody adhesion inhibited Zap70 reporter dynamics (*Tischer and Weiner, 2019*). With no additional adhesion, only actively signaling cells adhere to the bilayer. In the absence of adhesion beyond LOV2, we potentially missed sampling high occupancy, short half-life regimes with little Zap70 recruitment, as those cells failed to adhere to the bilayer. If we filter out the short half-life conditions of our new Zap70 data (with ICAM-1 adhesion) and refit our kinetic proofreading model, we measure a similarly low proofreading result as we previously observed in the absence of adhesion beyond LOV2 (n=0.7 ± 0.3) (*Figure 4—figure supplement 2*). With ICAM-1 adhesion we likely capture a more complete dataset of Zap70 recruitment, improving our measurement of upstream proofreading.

## Evidence for kinetic proofreading steps between Zap70 recruitment and LAT clustering, with further steps between LAT clustering and DAG generation

Next, we measured the strength of kinetic proofreading at the levels of LAT clustering and DAG generation. LAT clustering showed an increased dependency on binding half-life (ρ=0.54) and a decreased dependence on receptor occupancy (ρ=0.21) compared to Zap70 recruitment (*Figure 4—figure supplement 1B*). Our proofreading model fits our LAT clustering data with a higher degree of proofreading (n=7.8 ±1.1) compared to Zap70 recruitment (*Figure 4B*). The stronger degree of kinetic proofreading at LAT clustering versus Zap70 recruitment suggests additional steps of kinetic proofreading between the recruitment of Zap70 to phosphorylated ITAMs and the formation of LAT clusters. The generation of DAG also depended heavily on ligand half-life (ρ=0.52), while depending the least on occupancy (ρ=0.18) (*Figure 4—figure supplement 1C*). Our model fits the highest degree of kinetic proofreading at DAG generation (n=11.3 ± 1.5), suggesting further kinetic proofreading steps beyond initial LAT clustering and upstream of DAG generation (*Figure 4C*). The sequential increase in the degree of kinetic proofreading progressing down the antigen signaling cascade suggests

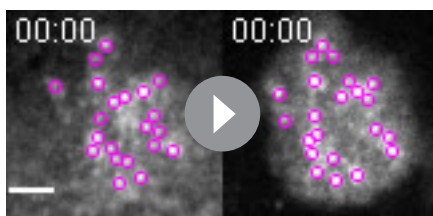

**Video 5.** Lifetime of LOV2 and Zap70 after optogenetic signal termination. Example cell from time course of bound LOV2 (left) and recruited Zap70 (right) loss after inactivation of antigen signaling with strong blue light. Jurkat cells expressing zdk-CAR and Zap70 biosensor were activated on LOV2 and ICAM-1 functionalized supported lipid bilayers (SLBs) for 3 min prior to the start of the video. Cells are illuminated with strong blue light at the start of the video, and the lifetime of bound LOV2 and recruited Zap70 are tracked using the ImageJ TrackMate plugin (magenta circles, MM:SS timestamp top left). Only tracks originating before blue-light illumination (start of the video) were included to filter out spurious tracks later in the time course (scale bar = 5µm).

https://elifesciences.org/articles/75263/figures#video5

steps contributing to kinetic proofreading exist throughout the cascade, at least up to the generation of DAG (*Figure 4D*).

To assess our model fits, we evaluated the residuals of each model subtracted from their respective dataset. For Zap70 recruitment, our model underestimates the degree of activation at moderate binding half-lives and receptor occupancies, as indicated by the positive region in the center of the heatmap. It is possible that Zap70 recruitment reaches saturation at shorter ligand-binding half-lives than our model predicts (*Figure 4—figure supplement 3A*). For both LAT clustering and DAG generation, our models performed poorest in the region of lowest occupancy and shortest half-life (*Figure 4—figure supplement 3B,C*). In this region of our dataset, the fluorescent signal from bound LOV2 above the background fluorescence of unbound LOV2 is smallest. To compensate for fluorescence of unbound LOV2, we subtract off the local background fluorescence of unbound LOV2 around each cell. In doing so we may be underestimating the amount of LOV2 bound to each cell, leading to an underestimation of signaling output by the models. Future studies at LOV2 densities approaching single molecule would better capture this regime of receptor occupancy, but cell-to-cell variation in activation would be too high to be compatible with our current steady-state analysis (*Lin et al., 2019*).

To globally assess how well our fit N values captured the datasets in comparison to other possible N values, we computed the average residual for models fit over a range of constant N values for each dataset (*Figure 4—figure supplement 3D*). The residual landscape for Zap70 shows a significant minimum around models with four or five proofreading steps, suggesting that these models perform substantially better than models with greater or fewer steps. The residual landscapes for LAT and DAG are much flatter, suggesting models with a greater or fewer numbers of upstream proofreading steps perform only marginally worse than our best fit models of N=7.8 and N=11.3 for LAT and DAG, respectively.

In light of the flat residual landscapes for LAT clustering and DAG generation, we used Akaike information criterion (AIC) to determine if models where LAT and DAG have the same number of upstream proofreading steps (same N value) better explained our data than our current model where Zap70, LAT, and DAG all have unique N values (*Yang, 2019*). The AIC-based selection process aims to balance model complexity with model fit in order to avoid overfitting and find the simplest model that accurately describes the data. Models with unique N values for all three signaling steps have the lowest AIC, thus best explaining our data. However, models where LAT and DAG share the same N value (and Zap70 has a unique N value) have only marginally worse AIC (*Figure 4—figure supplement 5*). This is expected if the LAT and DAG N values are indeed different, but the difference is small in magnitude. As the number of upstream proofreading steps increases, the transition of a signaling step from off to on becomes increasingly switch-like as a function of average ligand-binding half-life (*Figure 3D*, right). Confidently delineating between signaling steps in this switch-like regime will likely require finer control of binding half-lives than our current system's control of the average binding half-life.

In our previous work our model fit fewer (N=2.7) steps to DAG generation. We now fit a higher number of steps (N=11.3) to DAG generation. This change could be due to the incorporation of ICAM into our current study, which has been shown to potentiate ligand discrimination (*Pettmann et al., 2021*). Furthermore, our previous antibody-based adhesion may have short-circuited some proofreading steps by irreversibly holding the cell membrane close to the supported lipid bilayer. To

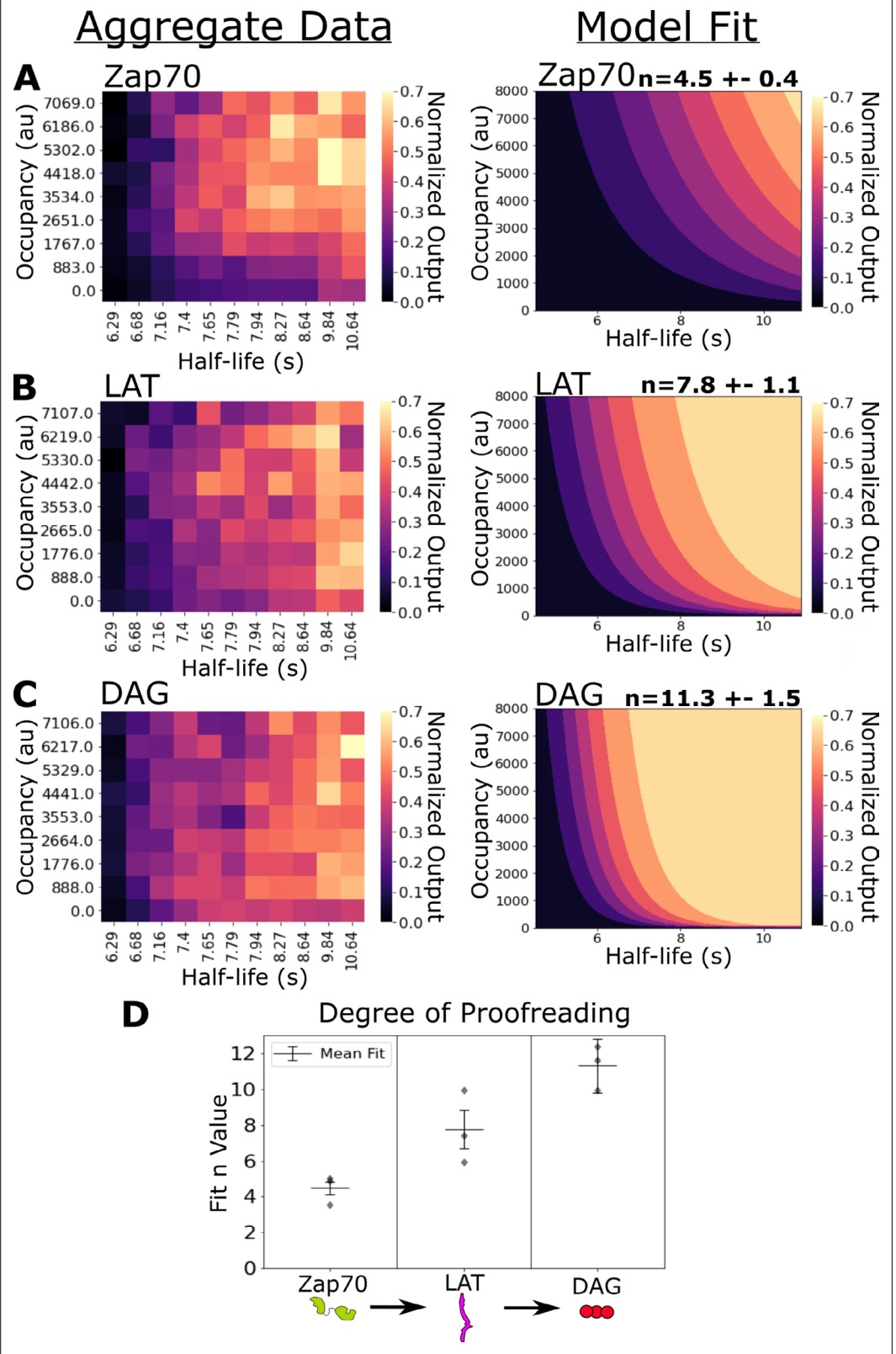

**Figure 4.** Kinetic proofreading starts upstream of Zap70 recruitment and progressively increases in strength at LAT clustering and diacylglycerol (DAG) generation. (**A**) Collected data of Zap70 recruitment as a function of receptor occupancy (y-axis) and ligand half-life (x-axis) (left). Zap70 recruitment best fits a model of kinetic proofreading with a strength of upstream proofreading of n=4.5 ±- 0.4 (right). (**B**) Collected data of LAT clustering

*Figure 4 continued on next page*

*Figure 4 continued*

as a function of receptor occupancy (y-axis) and ligand half-life (x-axis) (left). LAT clustering best fits a model of kinetic proofreading with a strength of upstream proofreading of n=7.8 ± 1.1 (right). (**C**) Collected data of DAG generation as a function of receptor occupancy (y-axis) and ligand half-life (x-axis) (left). DAG generation best fits a model of kinetic proofreading with a strength of upstream proofreading of n=11 ± 1.5 (right). (**D**) The mean model fit n-values (±1 std) for each biosensor across three biological replicates (gray markers). The fit n-values continue to increase from Zap70 recruitment to LAT cluster to DAG generation. These results suggest the existence of kinetic proofreading steps between ligand-binding and Zap70 recruitment, between the recruitment of Zap70 and the formation of a LAT cluster, and between the initial formation of a LAT cluster and the generation of DAG from that cluster.

The online version of this article includes the following source data and figure supplement(s) for figure 4:

**Source data 1.** Data tables (.csv) of the biological replicates used to fit the degree of kinetic proofreading (**n**) for each biosensor.

**Figure supplement 1.** Correlation of signaling output with receptor occupancy and ligand-binding half-life.

**Figure supplement 2.** Zap70 data subset replicates previously reported kinetic proofreading value.

**Figure supplement 3.** Residuals of kinetic proofreading model fits.

**Figure supplement 4.** Output of steady-state model from *Ganti et al., 2020*.

**Figure supplement 5.** Kinetic proofreading model selection using Akaike information criterion (AIC).

evaluate if our higher value fits are indeed the best fit values for our datasets, we fit our model to each dataset while holding the value of N constant in the range of 0–14 steps, and evaluated the average residual value for each model fit (*Figure 4—figure supplement 3D*). For all signaling steps, the fit value of N was near the minima of average residual and had a lower average residual value than a model with three proofreading steps.

To assess the plausibility of a larger number of proofreading steps, we implemented the steady-state kinetic proofreading model from *Ganti et al., 2020*. The model estimates the minimum number of proofreading steps required to discriminate between cognate-ligands and self-ligands with different binding half-lives present at a given concentration ratios at a given Hopfield error rate (*Hopfield, 1974*). First, we evaluated what combinations of ligand half-lives and concentration ratios an 11-step kinetic proofreading network could discriminate at an error rate less than $10^{-3}$ (*Figure 4—figure supplement 4A*). We chose the error rate of $10^{-3}$, as it is an order of magnitude less than the theorized $10^{-4}$ upper limit error rate of the native TCR (*Ganti et al., 2020*). At moderate half-life ratios, an 11-step network can discriminate cognate peptides present in small concentrations (e.g. 1 cognate-ligand per 1000 self-ligands at a half-life ratio of 6).

In our optogenetic system, the ratio of the average ligand-binding half-life between the longest suppressive half-life and the shortest fully activated half-life is about 2. However, an 11-step network is insufficient to discriminate between ligands with a half-life ratio of 2, even at the high ligand ratio of 1 (equal concentrations of cognate- and self-ligand). This suggests that our cells are unlikely to be detecting the average ligand-binding half-life of each blue-light condition, but are more likely detecting longer-lived binding events from the underlying distribution of half-lives. Another possibility is that our in vitro washout measurements, which measure average ligand-binding half-lives of soluble ligands diffusing in three dimensions, differ from the half-lives of ligand-receptor interactions between the cell's plasma membrane and the supported lipid bilayer diffusing in two dimensions (*Huang et al., 2010*).

To better explore the kinetic proofreading model space, we generated heatmaps reporting the required number of steps to discriminate combinations of ligand and half-life ratios at an error rate of $10^{-3}$ (*Figure 4—figure supplement 4B*). To discriminate between ligands with a half-life ratio of 2, at least 14 steps are needed when the ligands are at equal concentrations, and more than 25 steps are needed if cognate-ligands are 1 per 1000 self-ligands. The required number of proofreading steps decreases rapidly as the half-life ratio increases, reaching a minimum of eight steps needed for a concentration ratio of 1 per 1000 and a half-life ratio of 10, which is more in line with physiological half-life ratios between agonist and non-agonist peptides (*Davis et al., 1998*).

After comparing our results with the Ganti model, this analysis suggests that our number of fit proofreading steps may be somewhat inflated as a function of our uses the average ligand-binding

half-lives of three-dimensional washout experiments in place of the two-dimensional single-molecule information T cells use to make activation decisions. However, the higher fit N values are more consistent with the required number of steps to discriminate ligands under more physiological conditions than our previous measurements of approximately three steps, which would not be expected to discriminate ligands with half-life ratio of 10 even at a ligand ratio of 1 (*Figure 4—figure supplement 4B*, right).

## LAT clusters reset more slowly than Zap70 clusters upon ligand disengagement

Kinetic proofreading requires all signaling intermediates to reset upon ligand unbinding. While it is often assumed signaling intermediates reset at similar rates, slower reset rates for downstream and terminal signaling intermediates could improve sensitivity without a great loss in specificity (*McKeithan, 1995*). Our zdk-CAR deactivates with blue light, giving us the unique ability to unbind all antigen-binding events synchronously. After measuring additional kinetic proofreading steps downstream of the activated receptor, we used our Zdk-CAR system to measure the off-rate of bound LOV2, recruited Zap70, and clustered LAT following acute antigen unbinding (*Figure 5A*).

After allowing cells to activate on LOV-AF488 and ICAM-1-HIS functionalized bilayers for 3 min, we acutely unbound all LOV2 ligands with intense blue light while imaging LOV2 and Zap70 or LAT biosensors. We manually segmented cells in ImageJ and tracked individual subcellular clusters of each biosensor using the TrackMate plugin (*Tinevez et al., 2017*) to sample their lifetime distributions (*Figure 5B*, *Video 5* and *Video 6*). We assumed the loss of each biosensor consisted of one or more Poisson steps. A single-step mechanism results in exponentially distributed lifetimes, while a multistep process creates an Erlang lifetime distribution, where the integer shape parameter estimates the number of steps (*Huang et al., 2019*). We found the lifetimes of bound LOV2 and recruited Zap70 best fit exponential distributions with mean lifetimes of 3.3 and 9.3 s, respectively. However, the lifetimes of LAT clusters best fit a two-step Erlang distribution (shape = 2) with a mean lifetime of 14.2 s (*Figure 5C*). Upon ligand unbinding, recruited Zap70 and clustered LAT likely reset through different processes, with LAT resetting through a slower multistep process. While this idea awaits experimental verification, these data open the possibility that a LAT cluster could survive a momentary ligand unbinding event (loss of Zap70) and survive long enough to integrate multiple ligand-binding events, as observed in primary T cells on SLBs with native ligand (*Lin et al., 2019*). Such a mechanism could explain why our simple steady-state model of kinetic proofreading captures Zap70 recruitment better than LAT clustering and DAG generation, as our ligand density and stimulation time are likely sufficient for binding event integration to occur (*Figure 4—figure supplement 3*).

## Discussion

Kinetic proofreading enables T cells to discriminate self- from cognate-antigens by converting small changes in antigen-binding half-life to large changes in cell activation. Amplifying small changes in binding half-life requires a chain of multiple slow (relative to binding half-life) biochemical events that separate short binding events from long binding events. Where these events exist in the antigen signaling cascade is not fully understood.

In this study, we improved our optogenetic proofreading assay with ICAM-1 adhesion and measured the dependency of Zap70 recruitment, LAT clustering, and DAG generation on antigen-binding half-life and receptor occupancy. We now find evidence for kinetic proofreading in antigen signal transduction as early as Zap70 recruitment to phosphorylated receptor ITAMs. We measured greater kinetic proofreading at the clustering of the scaffold protein LAT, and the highest degree of kinetic proofreading at the generation of the signaling lipid DAG (*Figure 4D*). We also measure the reset rate of Zap70 and LAT upon ligand unbinding. Our findings suggest that the kinetic proofreading underlying T cell antigen discrimination spans multiple membrane-associated signaling complexes (*Yi et al., 2019*).

After a receptor binds an antigen, many events must occur before Zap70 is recruited to the receptor's ITAMs. Many strong candidates exist for the mediators of this kinetic processing (*Chakraborty and Weiss, 2014*). Ligand binding excludes the bulky phosphatase CD45 that would otherwise rapidly dephosphorylate the ITAMs (*Davis and van der Merwe, 2006*; *Springer, 1990*).

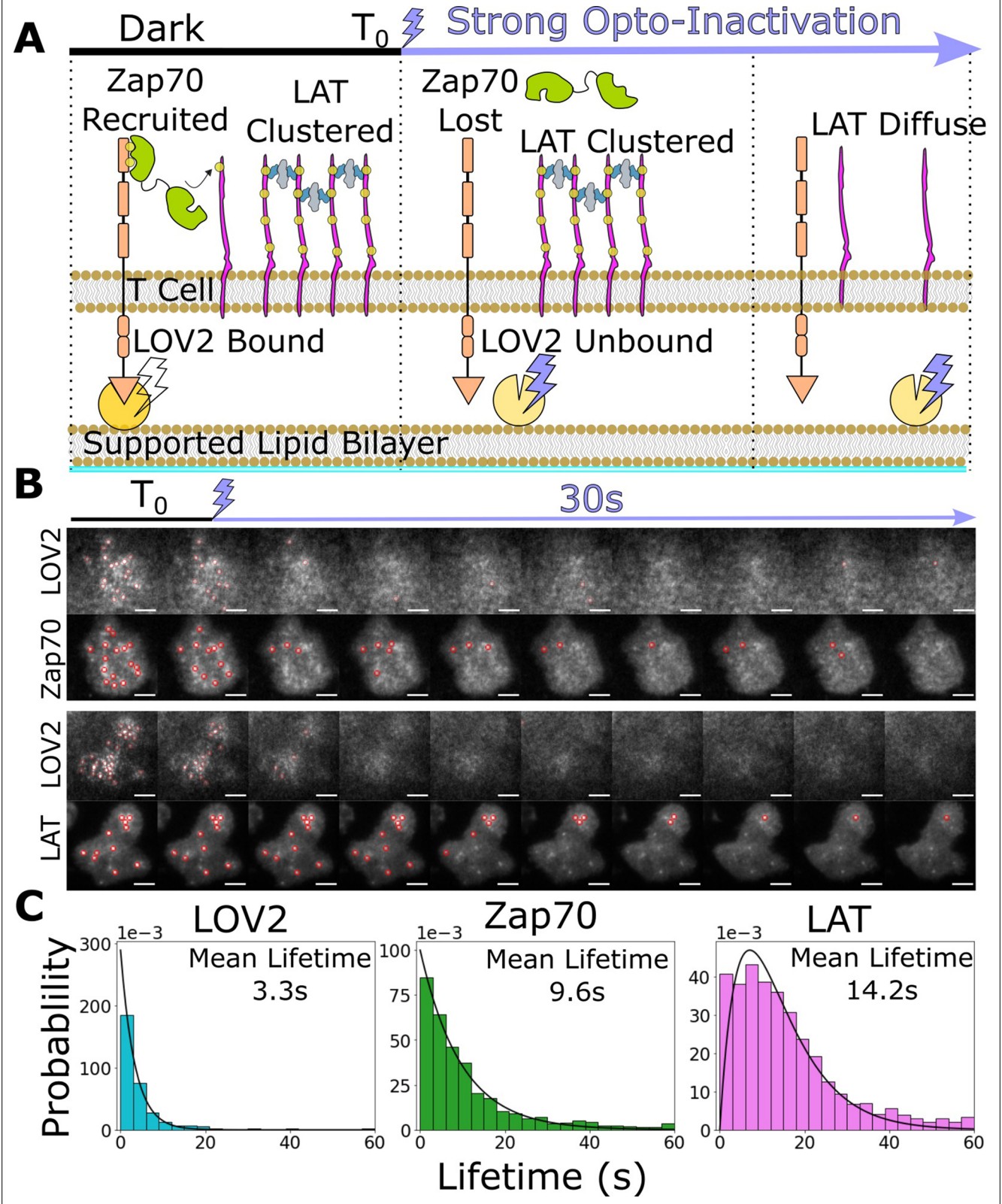

**Figure 5.** Downstream signaling complexes reset slower than active receptors upon ligand unbinding. (**A**) Schematic of reset experiment. After synchronously unbinding all receptors with intense blue light, we measure the reset rate of LOV2 unbinding, Zap70 loss, and LAT cluster dissociation. (**B**) Representative cells of Zap70 and LAT upon receptor inactivation with respective LOV2 images. Cells expressing either Zap70-Halo (top) or LAT-Halo (bottom) were allowed to activate for 3 min before acute inactivation with intense blue light for 1 min and subsequently imaged every 3 s. The

*Figure 5 continued on next page*

Figure 5 continued

lifetimes of subcellular clusters of LOV2, Zap70, and LAT were tracked with the ImageJ TrackMate plugin (scale bar = 5μm). (C) The lifetime distributions of tracked clusters after receptor inactivation. The lifetime distributions of LOV2 unbinding and Zap70 dissociation fit exponential distributions with mean lifetimes of 3.3 and 9.3 s, respectively, suggesting a one-step reset process. The LAT declustering lifetime distribution fit an Erlang distribution of a two-step process and a mean lifetime of 14.2 s. Fitting of cluster intensity over time with exponentials gave comparable results for all three reporters (*Figure 5—figure supplement 1*).

The online version of this article includes the following source data and figure supplement(s) for figure 5:

**Source data 1.** Data tables (.csv) for each cell used for the lifetime distributions of each reporter.

**Figure supplement 1.** Bi-exponential fit of biosensor intensity reset curve is comparable to lifetime analysis from *Figure 5*.

The receptor itself could undergo mechanical conformational changes making it more accessible for phosphorylation (*Kim et al., 2009*; *Kim et al., 2010*). An active molecule of the src-family-kinase Lck must phosphorylate the receptor (*Schoenborn et al., 2011*; *Tan et al., 2014*). Finally, Zap70 must overcome autoinhibition before binding the phosphorylated receptor with both of its tandem SH2 domains (*Deindl et al., 2009*; *Hsu et al., 2017*). Additional proofreading steps may also contribute that are not captured in our system such as CD4/8 coreceptor scanning (*Stepanek et al., 2014*) and CD3ε ITAM allostery (*Borroto et al., 2014*). Others have also suggested receptor-level proofreading through measuring a temporal delay between receptor binding and Zap70 recruitment for both TCRs (*Huse et al., 2007*; *Yi et al., 2019*) and CARs (*Bhatia et al., 2005*).

We measured stronger proofreading downstream of Zap70 recruitment for the clustering of LAT and the generation of DAG. This result implies that additional kinetic proofreading steps exist separate from the receptor. The additional proofreading steps measured at LAT clustering could include the full activation of Zap70 through phosphorylation and the many phosphorylation and binding events on LAT required to initiate clustering. Hyperactive Zap70 mutants biased toward the active conformation increase T cell response to normally weak agonists, suggesting that pre-activation of Zap70 short-circuits one or more proofreading steps (*Shen et al., 2021*).

While our models are less confident about additional kinetic proofreading in DAG generation, many potential kinetic proofreading steps exist beyond the initial clustering of LAT. LAT does not require the phosphorylation of all its tyrosine residues, or binding of all known associated proteins, to cluster (*Su et al., 2016*). Further phosphorylation and protein recruitment events may be required for an existing LAT cluster to mature into a fully signaling-competent complex. A strong candidate for an additional proofreading step is phosphorylation of LAT Y132. Y132 is an evolutionarily conserved poor substrate for its kinase Zap70, and this phosphorylation site is required for PLCγ recruitment, DAG generation, and Ca++ mobilization (*Andreotti et al., 2010*; *Courtney et al., 2018*). Mutation of the residues around Y132 to a better Zap70 substrate that is more rapidly phosphorylated resulted in decreased T cell antigen discrimination (i.e. strong activation to weak agonists; *Lo et al., 2019*). Furthermore, *Sherman et al., 2011* found that PLCγ is recruited to only a subset of LAT clusters, suggesting that only a subset of LAT clusters reach a fully signaling-competent state.

The kinetic proofreading model requires all intermediate steps to reset upon unbinding of the ligand (*Figure 1A*). This means that information about the receptor's binding state must be communicated to all proofreading steps. If kinetic proofreading steps exist beyond the T cell receptor, how is unbinding information conveyed to these effectors? Importantly, there is evidence of physical proximity of LAT with the receptor. While TCR/Zap70 and LAT/PLCγ microclusters

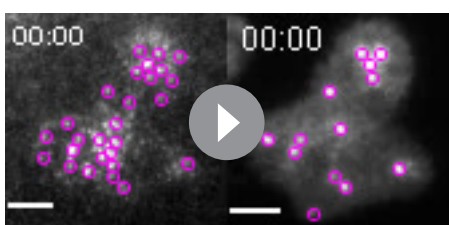

**Video 6.** Lifetime of LOV2 and LAT after optogenetic signal termination. Example cell from time course of bound LOV2 (left) and LAT clusters (right) loss after inactivation of antigen signaling with strong blue light. Jurkat cells expressing zdk-CAR and LAT biosensor were activated on LOV2 and ICAM-1 functionalized supported lipid bilayers (SLBs) for 3 min prior to the start of the video. Cells are illuminated with strong blue light at the start of the video, and the lifetime of bound LOV2 and recruited Zap70 are tracked using the ImageJ TrackMate plugin (magenta circles, MM:SS timestamp top left). Only tracks originating before blue-light illumination (start of the video) were included to filter out spurious tracks later in the time course (scale bar = 5μm).
https://elifesciences.org/articles/75263/figures#video6

form spatially segregated domains, these domains remain adjacent to one another (*Yi et al., 2019*). *Lo et al., 2018*, demonstrated that the protein Lck binds Zap70 with its SH2 domain and LAT with its SH3 domain, potentially bridging the two signaling domains together and propagating binding information.

An attractive reset mechanism is the segregation of CD45 away from bound receptors, creating spatial regions in which TCR- and LAT-associated activating events can occur (*Davis and van der Merwe, 2006*). Super-resolution microscopy by *Razvag et al., 2018* measured TCR/CD45 segregated regions within seconds of antigen contact at the tips of T cell microvilli. Upon unbinding, these regions of phosphatase exclusion collapse, allowing CD45 to dephosphorylate receptor ITAMs and LAT clusters. However, the rate of dephosphorylation for LAT and receptor ITAMs could differ. LAT clusters exclude CD45 in reconstituted bilayer systems, potentially limiting the dephosphorylation to LAT molecules at the edges of the cluster thus slowing reset (*Su et al., 2016*). The kinetics of multivalent protein-protein interactions within TCR and LAT clusters can also influence dephosphorylation and dissociation rates (*Goyette et al., 2022*). A CD45-based mechanism could also reset other proofreading signaling pathways occurring in the phosphatase excluded zone, such as the activation of SOS which *Huang et al., 2019*, recently showed exhibits kinetic proofreading behavior.

What are the advantages of distributing kinetic proofreading beyond the receptor? Having subsequent proofreading steps separate from the receptor potentially allows cooperativity between multiple receptors in activating downstream signaling steps (e.g. multiple activated receptors contributing to the creation and maturation of a single LAT cluster). Consistent with this idea, larger TCR cluster size increases the probability of T cell activation (*Manz et al., 2011*). *Lin et al., 2019*, also observed cooperativity between single TCR/pMHC-binding events, where shorter binding events that overlapped in space and time activated NFAT translocation similar to long-lived single binding events in primary murine T cells.

Work on kinetic proofreading often assumes that the reset rate after ligand binding is uniform across all signaling intermediates ($k_{off}$, *Figure 1A*). However, in his seminal work applying the mathematical framework of kinetic proofreading to TCR activation, Timothy McKeithan postulated that a kinetic proofreading system with slower reset rates for downstream events compared to upstream signaling events would respond to true cognate-antigen-binding events a higher percentage of the time while still ignoring weak self-antigen-binding events (*McKeithan, 1995*). Recently, Pettman et al. modeled how slower resetting of terminal signaling complexes could account for fractional steps in their measurements of the number of proofreading steps (*Pettmann et al., 2021*). We measured slower dissociation of downstream LAT clusters through a slow multistep process compared to the upstream events of ligand binding and Zap70 recruitment which followed a more rapid single-step reset process (*Figure 5C*). Our data opens the possibility that LAT clusters could survive transient unbinding events that deactivate an active receptor by being rescued by other transiently active receptors, resulting in a fully mature signaling complex created by the integration of multiple shorter binding events and potentially improving ligand discrimination as McKeithan proposed. It remains to be seen if individual LAT clusters can be rescued after ligand unbinding, or if a LAT cluster is destined to be fully reset once the binding event that led to its creation ends.

Recently, Harris et al. quantified the reset rate of the downstream signaling events Ca++ release and ERK phosphorylation upon signal inhibition to be 29 s and 3 min, respectively (*Harris et al., 2021*). They showed both Ca++ and ERK levels can persist across short inhibitions of signaling. What makes LAT clusters different than these persistent downstream events? The dissolution of LAT clusters is directly triggered by the unbinding of ligand from the TCR, and both the TCR and LAT are de-phosphorylated by CD45. To our knowledge, the rate of ERK dephosphorylation or cytosolic Ca++ depletion is not accelerated by TCR unbinding, and is turned over through constant rather than agonist-gated degradation. A useful future line of inquiry would be to quantify the reset rate for signaling steps throughout the cascade upon ligand unbinding versus orthogonal signal inhibition (e.g. kinase inhibition).

Cooperativity between active receptors to complete downstream proofreading steps could explain observations of T cell activation in response to intermediate affinity self- and tumor antigens (*Yin et al., 2012*). Intermediate lifetime-binding events could last long enough to complete all necessary steps for receptor activation but not long enough to finish proofreading steps downstream of the receptor. If enough intermediate binding events overlapped in space and time, they could cooperate in accomplishing downstream proofreading events, either through the acceleration of signal

propagation through the cascade or through mutual prevention of system reset upon unbinding of a subset of ligands. Such a cooperativity mechanism has been implicated in autoimmunity (*Korem Kohanim et al., 2020*; *Wang et al., 2020*) and immunotherapy off-target effects (*Cameron et al., 2013*).

## Materials and methods

Methods were adapted from our lab's previous work (*Tischer and Weiner, 2019*) with modified sections highlighted below.

### Cloning

We used standard molecular biology protocols for all cloning. In general, we PCR-amplified individual DNA segments and assembled them using the isothermal Gibson assembly method. Mark M Davis (*Huse et al., 2007*) kindly gifted us a plasmid encoding the C1 domains of PKCθ. Jay Groves kindly gifted us plasmids encoding human Zap70 and LAT (*O'Donoghue et al., 2013*). The Zdk-CAR was based on a CD8-CAR (*Irving and Weiss, 1991*), the plasmid for which was a kind gift from Art Weiss. The V529N mutation in LOV2 biases it toward the 'open' conformation which does not bind Zdk (*Yao et al., 2008*). This mutation facilitated the quick release of LOV2 from the Zdk-CAR.

### Cell culture

Jurkat cells grew in RPMI 1640 (Corning Cellgro, #10-041-CV) supplemented with 10% fetal bovine serum (Gibco, #16140-071) and glutamine (Gibco, #35050-061). Jurkat lines were tested to be negative for mycoplasma. Cell identity (Jurkat E6.1 clone) was validated with STR profiling. We maintained Jurkats at densities between 0.1 and $1.0 \times 10^6$ cells per ml. We grew 293T cells in DMEM (Gibco, #11995-065) with 10% fetal bovine serum. All cell lines grew in humidified incubators at 37°C with 5% $CO_2$.

### Cell line construction

We combined 1 ml of wild-type (WT) Jurkat cells at $0.5 \times 10^6$ cells/ml with 0.5 ml lentiviral supernatant for the zdk-CAR and the appropriate reporter construct. Cells recovered overnight in the incubator, 8 ml of media was added the following day and cells were grown to desired density. Cells were labeled with the Halo dye JF549 (*Grimm et al., 2015*) and we sorted the desired expression levels by FACS (FACSAria II, BD). Cells recovered for approximately three passages and tested for blue-light-dependent signaling on LOV2 and ICAM-1 functionalized SLBs on the microscope. We obtained WT Jurkat cells for this study from the laboratory of Dr Art Weiss. Regular mycoplasma tests were negative.

### Lentiviral production

We produced lentivirus in 293T cells using a second-generation lentiviral system (*James and Vale, 2012*). We transfected cells grown to 40–60% confluency in six-well plates with 0.5 µg each of pHR (containing the transgene of interest), pMD2.G (encoding essential packaging genes), and p8.91 (encoding VSV-G gene to pseudotype virus) using 6 µl of Trans-IT (Mirus, #MIR 2705) per manufacturer's instructions. (Plasmids kind gift from Ron Vale.) After 48 hr, we filtered the supernatant through a 0.22 µm filter and used immediately or froze at −80°C until use.

### Cell preparation for imaging

For each imaging well, we used approximately $1 \times 10^6$ Jurkat cells labeled with the Halo dye JF549 (*Grimm et al., 2015*) (10 nM, a kind gift from the Lavis lab) for at least 15 min at 37°C before resuspension in growth media. Before imaging, we washed cells once into mHBSS-BB (at 400 RCF, 4 min), and resuspended them in 40 µl imaging media before adding them to a functionalized SLB.

### Protein purification

LOV2 and Zdk were purified and labeled as described in our previous work (*Tischer and Weiner, 2019*). Jay Groves generously gifted us the purified human ICAM-1-HIS used in this study (*Nye and Groves, 2008*).

## Glassware cleaning

All glassware was cleaned as described in our previous work (*Tischer and Weiner, 2019*).

## Preparation of small unilamellar vesicles

We washed a precleaned 4 ml glass vial ×2 with chloroform (Electron Microscopy Sciences, #12550). Using Hamilton syringes (Hamilton Company, Gastight 1700 series, #80265 and #81165), we combined 1 mmoles of lipids in the following molar ratio: 97.5% DOPC, 0.5% PEG-PE, 1% DGS-NTA(Ni), and 1% biotinyl CAP PE (Avanti Polar Lipids, #850375C, #880230C, #790404C, #870282C, respectively). Next, we evaporated the chloroform by slowly rotating the vial at an angle while slowly flowing nitrogen gas (Airgas, #NI 250). We vacuum desiccated the resulting lipid film for 2 hr to overnight. After desiccation, we rehydrated the lipids in 2 ml of TBS and gently vortexed the vial for 10 min. We transferred the mixture to a cleaned 5 ml round bottom glass tube. We formed small unilamellar vesicles (SUVs) by submerging bottom of the tube in a Branson 1800 ultrasonic cleaner (Branson #M1800) to the level of the lipid mixture in the tube and sonicating for 30–60 min until clear. We added ice periodically to the sonicator bath to maintain a temperature of 0°C. Centrifugation at >21,000 RCF for 30 min at 4C (Eppendorf microcentrifuges 5425R) pelleted large lipid structures. We removed the SUV containing supernatant and used it immediately or stored it in liquid nitrogen until use.

## RCA cleaning of microscopy coverslips

We placed glass coverslips (Ibidi, #10812) into a glass Coplin jar (Sigma-Aldrich, #BR472800) and successively bath sonicated for 10 min each in acetone (Sigma-Aldrich, #534064-4L), isopropyl alcohol (Thermo Fisher Scientific, #BP2618500), and ddH$_2$O. Coverslips were washed five times in ddH$_2$O between each bath sonication to remove excess organic solvents. Next, we added 40 ml ddH$_2$O, 10 ml of 30% ammonium (Thermo Fisher Scientific, #423305000) hydroxide, and 10 ml 30% hydrogen peroxide (VWR, #7722-84-1) to the coverslips. We placed the Coplin jar into a 70–80°C water bath and allowed it to react for 10 min after the base solution began to vigorously bubble. We decanted the base solution and washed the coverslips five times in ddH$_2$O. Next, we added 40 ml ddH$_2$0, 10 ml of 30% hydrochloric acid (Millipore Sigma, #1003180250), and 10 ml 30% hydrogen peroxide to the coverslips. Again we incubated the reaction in the water bath and allowed it to react for 10 min after the acid solution began to vigorously bubble. We decanted the acid solution and washed the coverslips five times in ddH$_2$O and stored them in ddH$_2$O for up to 1 week.

## Functionalization of SLBs and cell preparation

After removing an RCA cleaned glass coverslip from ddH$_2$O and immediately blown drying it with compressed nitrogen, we firmly attached a six-well Ibidi sticky chamber (Ibidi, #80608) to the coverslip. We diluted 30 μl of SUV mixture with 600 μl TBS before adding 100 μl to each well and incubated at room temperature for 25 min. To functionalize a well, we flushed out excess lipids with 500 μl TBS. Next, we added 100 μl of ICAM-1-HIS diluted in TBS-BB to 150 μM to the well and incubated at room temperature for 35 min. After incubation, we washed the well with 500 μl TBS before adding 100 μl Streptavidin (Rockland, #S000-01) diluted in TBS-BB (2 μg/ml final) to the well and incubated at room temperature for 5 min. After washing again with 500 μl TBS-BB, we added LOV2 diluted in TBS-BB (typically between 20 and 200 nM) to the well and incubated in the dark at 37°C for 5 min. We then flushed the well with 500 μl HBSS-BB and incubated with cells previously labeled with the halo dye washed into Imaging media. Cells adhered to the SLB in the dark for 5 min at 37°C before imaging.

## Buffers for SLB functionalization and imaging

- TBS: 150 mM NaCl, 20 mM Tris Base, pH 7.5
- TBS-BB: TBS with 2 mg/ml BSA and 0.5 mM βME.
- mHBSS: 150 mM NaCl, 40 mM KCl, 1 mM CaCl$_2$, 2 mM MgCl$_2$, 10 mM glucose, 20 mM HEPES, pH 7.2
- mHBSS-BB: mHBSS with 2 mg/ml BSA and 0.5 mM βME
- Imaging media: mHBSS-BB supplemented with 2% fetal bovine serum, 50 μg/ml ascorbic acid and 1:100 dilution of ProLong Live Antifade Reagent (Thermo Fisher Scientific, #P36975). Solution incubated at room temperature for at least 90 min to allow the antifade reagent to reduce oxygen levels.

## Microscopy

Imaging was performed on an Eclipse Ti inverted microscope (Nikon) with two tiers of dichroic turrets to allow simultaneous fluorescence imaging and optogenetic stimulation. The microscope was also equipped with a motorized laser TIRF illumination unit, ×60 and ×100 Apochromat TIRF 1.49 NA objective (Nikon), an iXon Ultra EMCCD camera (Andor), and a laser launch (Versalase, Vortran) equipped with 405, 488, 561, and 640 nm laser lines. For RICM, light from a Xenon arc lamp (Lambda LS, Sutter Instrument) source was passed through a 572/35 nm excitation filter (Chroma, #ET572/35x) filter and then a 50/50 beam splitter (Chroma, #21000). Microscope and associated hardware was controlled with MicroManager (*Edelstein et al., 2014*) in combination with custom-built Arduino controllers (Advanced Research Consulting Corporation). Blue light for optogenetic stimulation was from a 470 nm LED (Lightspeed Technologies Inc, #HPLS-36), controlled with custom micromanager scripts. For most timepoints, only RICM and TIRF561 images were collected. During and in-between these timepoints, a TIRF488 long-pass dichroic mirror remained permanently in the top dichroic turret, ensuring the blue-light illumination of the cells was never interrupted. The top TIRF488 dichroic passed the longer wavelengths used for RICM and TIRF561. Only when LOV2 localization was imaged with TIRF488 at the end of a 3 min stimulation was the top dichroic removed to allow the shorter fluorescence excitation light to pass.

## Image processing

After each day of imaging, we captured TIRF488, TIRF561, and RICM images of slides with concentrated solutions of fluorescein, Rose Bengal, or dPBS, respectively. To flat field correct, we subtracted the camera offset from both the experimental and dye images. By dividing the experimental image by the median dye slide image, we acquired the final flat field corrected image used in analysis (*Model, 2006*).

## Time course overview

We exposed cells to 5 min blocks of constant blue-light stimulation. Each block consisted of an initial 2 min hold in strong blue light followed by a 3 min stimulation at a fixed intensity of intermediate blue light. To measure the reporter output, we averaged the final four timeframes of each condition. We measured CAR occupancy from a long exposure TIRF488 image taken at the end of the 3 min stimulation, after the last biosensor output measurement made in TIRF561. As fluorescence excitation light from TIRF488 potently stimulates LOV2, the TIRF488 channel could only be imaged once at the very end of a 3 min stimulation. We repeated 5 min blocks over the course of an hour experiment with a variety of blue-light intensities to sample different ligand-binding half-lives.

## LOV2-binding half-life measurements

We calculated the average Zdk-binding half-life for each blue-light condition as described previously (*Tischer and Weiner, 2019*).

## CAR occupancy measurements

We background subtracted and thresholded RICM images at each timeframe to create a mask of cell footprints. The thresholded image was labeled using Python skimage watershed algorithm (*van der Walt et al., 2014*). A second local background mask was made by labeling pixels surrounding an expanded perimeter of each labeled cell in the cell mask. At steady state, free LOV2 should be homogeneously distributed on the SLB. The mean TIRF488 pixel intensity of a cell footprint is the sum of free LOV2 and LOV2 bound to the CAR. The mean TIRF488 pixel intensity in the background mask reflects free LOV2. Therefore, we calculated CAR occupancy as the mean TIRF488 pixel intensity in the cell mask minus the mean TIRF488 pixel intensity in the background mask.

## Biosensor measurements

To calculate biosensor output levels at steady state, we averaged the TIRF561 pixel intensity within each labeled cell mask over the last four timeframes (equivalent to the last 40 s) of a 3 min hold in blue light. To account for differences in biosensor expression level, we normalized cells to their average biosensor intensity in the absence of signaling (taken as the average of the last two TIRF561 images of each 2 min reset pulse of intense blue light). We sometimes observed drift in a cell's basal activity

over time. To correct for this drift, we subtracted the mean TIRF561 pixel intensity of each preceding 2 min reset pulse of intense blue light from the mean value of the proceeding 3 min stimulation of blue light. Thus, the reported biosensor output value is the fold-change from the cell's average basal activity minus the fold-change from resent basal activity.

Prior to model fitting, the biosensor output values of a biological replicate dataset (consisting of multiple wells acquired on the same day) were normalized by the 90th percentile output value in the dataset to properly normalize the data to the model, and to allow comparison between biosensors of variable dynamic ranges.

## Kinetic proofreading model fitting

We fit each biological replicate dataset to the simple model of kinetic proofreading described previously (*Figure 3D*; *Tischer and Weiner, 2019*). Each datapoint consists of an average ligand-binding half-life, a CAR occupancy measurement, and a biosensor output measurement (see above). With those three measured values, we fit the remaining three parameters of our model n, K, and β (number of proofreading steps, input for half-max biosensor output, and basal signaling output, respectively) using the Levenberg-Marquardt least-squares algorithm as implemented by the curve_fit function from the Python SciPy library (*Virtanen et al., 2020*).

## Akaike information criterion

Assuming errors are identically and normally distributed, AIC was calculated as 2K+n*ln(RSS/n) for several nested models where various conditions of equality were set on the N parameters in the equilibrium model across the data. K is the total number of parameters used to describe the outcomes, n is the total number of measurements across the system, and RSS is the sum of squared residual from the fit. The model with the lowest AIC score (three unique values for N) was chosen as the best model (*Yang, 2019*).

## Criteria for including or excluding cells in analysis

We restricted our analysis to cells that were present for the full-time course. Cells that detached partway through the time course or arrived after the experiment began were excluded. Each replicate began with a no-light (maximal half-life) condition to identify non-responding cells. Cells that did not exhibit at least a 10% increase in reporter output above their basal output were excluded from analysis.

## Biological and technical replicates

Kinetic proofreading datasets (*Figure 4*) – A biological replicate consisted of two time courses of stimulating cells with blue light on SLBs with different concentrations of LOV2, all on the same day (to ensure the light path did not change). We conducted each biological replicate on different days, with new preparations of cells, SLBs, and LOV2. Each time course within a biological replicate contained approximately 30 cells, whose biosensor output levels and receptor occupancy were measured in all half-life conditions. As the microscopy experiments could not be done in parallel and each biological replicate took an entire day, we could not conduct technical replicates. Data underlying each replicate included in the supplement.

Signaling reset datasets (*Figure 5*) – A biological replicate consisted of four cycles of acutely terminating signaling and measuring the respective reporter lifetimes for approximately 30 cells. For each reporter, three or four biological replicates were conducted with different preparations of cells and bilayers (data included in the supplement).

## Measuring signaling reset after ligand unbinding

OptoCAR expressing Jurkats were activated for 5 min on a LOV2-AF647 and ICAM-1 functionalized bilayer. At T=0, we illuminated the cell with intense blue light to unbind receptor-bound LOV2 ligands while imaging LOV2 and either Zap70 or LAT biosensors every 3 s. We manually segmented cells in ImageJ before tracking clusters of LOV2 and Zap70/LAT biosensors using the TrackMate plugin (*Tinevez et al., 2017*). We selected all tracks that persisted for at least two frames prior to blue-light illumination and plotted the lifetime distribution of the resulting tracks. We then fit each lifetime distribution to Erlang distributions with shape parameter k and rate parameter $\lambda$ (which reduces to

an exponential distribution when k=1) using curve_fit function from the SciPy library (*Virtanen et al., 2020*):

$$f(x) = \frac{\lambda^k x^{k-1} e^{-\lambda x}}{(k-1)!}$$

## Acknowledgements

We thank Art Weiss, Wan-Lin Lo, and Theresa Kadlecek for cell lines and thoughtful discussion, Jay Groves, Kiera Wilhelm, Mark O'Dair, and Nugent Lew for purified human ICAM-1-His, expertise in SLB preparation, help with LOV2 protein purification, and thoughtful discussion. Hana El-Samad, Kirstin Meyer, and Shohini Sen-Britain for helpful discussion and critical reading of the manuscript. This work was supported by the ARCS foundation (DB) National Science Foundation Predoctoral Fellowship (JPT), GM118167 (ODW), the NSF Center for Cellular Construction (DBI-1548297), and a Novo Nordisk Foundation grant for the Center for Geometrically Engineered Cellular Systems (NNF17OC0028176).

## Additional information

### Funding

| Funder | Grant reference number | Author |
| --- | --- | --- |
| National Institutes of Health | GM118167 | Orion David Weiner |
| National Science Foundation | DBI-1548297 | Orion David Weiner |
| Novo Nordisk Foundation Center for Basic Metabolic Research | NNF17OC0028176 | Orion David Weiner |
| National Science Foundation | Predoctoral Fellowship | Jason P Town |
| Achievement Rewards for College Scientists Foundation | Predoctoral Fellowship | Derek M Britain |

The funders had no role in study design, data collection and interpretation, or the decision to submit the work for publication.

### Author contributions

Derek M Britain, Conceptualization, Software, Funding acquisition, Investigation, Visualization, Methodology, Writing - original draft, Writing - review and editing; Jason P Town, Software, Investigation, Methodology, Writing - review and editing; Orion David Weiner, Conceptualization, Resources, Funding acquisition, Writing - original draft, Writing - review and editing

### Author ORCIDs

Derek M Britain http://orcid.org/0000-0002-3139-3797
Orion David Weiner http://orcid.org/0000-0002-1778-6543

### Decision letter and Author response

Decision letter https://doi.org/10.7554/eLife.75263.sa1
Author response https://doi.org/10.7554/eLife.75263.sa2

## Additional files

### Supplementary files
• Transparent reporting form

## Data availability

"Figure 4 - Source Data" and "Figure 5 - Source Data" files contain the numerical data used to generate the figures.

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
