## [Editor Report]

The authors have used an ontogenetic approach to advance understanding of the kinetic proofreading steps down-street of the antigen receptors. The authors have provided evidence for longer lifetimes of down-stream signaling events in relation to physiological receptor-ligand binding kinetics. This accounts for the ability of immune cells to respond across a large range of agonist pMHC while retaining the ability to prevent responses to very short lived interactions associated with self ligands. The results advance our knowledge of chimeric antigen receptor signaling and provides an approach to identify optimal receptor binding kinetics to couple to down-stream signaling machinery.

---

## [Decision Letter]

**Decision letter after peer review:**

Thank you for submitting your article "Progressive enhancement of kinetic proofreading in T cell antigen discrimination from receptor activation to DAG generation" for consideration by *eLife*. Your article has been reviewed by 2 peer reviewers, and the evaluation has been overseen by a Reviewing Editor and Aleksandra Walczak as the Senior Editor. The reviewers have opted to remain anonymous.

There is a consensus that your work is an advance on your prior study looking at ontogenetic control of T cell activation. The addition of the adhesion system enables a better analysis of how the kinetics of the light driven receptor-ligand interactions impact receptor proximal and distal signals. The work reinforces that early events fit within a framework of a kinetic proofreading (KP) model. Reviewers are concerned that later events are too slow in relation to receptor-ligand binding to support a kinetic KP process, unless the proofreading process incorporated down stream transcriptional and epigenetic steps in the model where early signals influenced by the receptor-ligand binding kinetic through KP controlled slower immune synapse levels on and off rates that then feed forward on the slower downstream events. This might be an interesting thing to discuss, but seems beyond the scope of the current modelling related to receptor-ligand binding processes and KP. The most obvious revision path would be to moderate claims as suggested by reviewers or to propose a set of plausible alternative models to KP as to how the later steps relate to the receptor-ligand binding kinetics and determine if KP is really the best model or if other models fit equally well or better at later stages.

1) An estimate of the goodness-of-fit is important to provide some confidence that the model used to fit the experimental data is sound. Alternative models might be fit just as well and might provide very different conclusions otherwise; the data fits to KP 'by default'. It would also be helpful to be more explicit in the methods about how this fitting was done. Could the authors model the data assuming a series of reversible reactions (ie not KP), with the rate constant for each being sufficiently low to still fit the delay observed between receptor engagement and downstream output. How would this compare to the KP model?

2) In the paragraph beginning with line 144: It should be noted that the kinetic proofreading model that the authors use is very simplified. The scaling with the power of steps, n, is only correct under certain approximations (see Hopfield's original PNAS paper). While the authors' note one of these approximations later (line 155), this is not the only one, and so mentioning the approximate nature of the analyses earlier on would be good.

3) As Figure 3 shows, there is quite a bit of cell-to-cell variability. Can this variability data be quantified to provide insights into this "noise", which is important in determining the response of the population of cells?

4) The authors may be interested in a paper by R. Ganti et al., (PNAS, 2020), which studies related issues using computational modeling, information theory, and experiments. In that paper, the consequences of "distributed" kinetic proofreading, a model for the reset steps along the pathway, and experiments considering some spatial aspects of kinetic proofreading are considered.

5) I think it would be appropriate to cite the original kinetic proofreading papers by Hopfield and Ninio.

*Reviewer #1 (Recommendations for the authors):*

1) In the paragraph beginning with line 144: It should be noted that the kinetic proofreading model that the authors use is very simplified. The scaling with the power of steps, n, is only correct under certain approximations (see Hopfield's original PNAS paper). While the authors' note one of these approximations later (line 155), this is not the only one, and so mentioning the approximate nature of the analyses earlier on would be good.

2) As Figure 3 shows, there is quite a bit of cell-to-cell variability. Can this variability data be quantified to provide insights into this "noise", which is important in determining the response of the population of cells?

3) The authors may be interested in a paper by R. Ganti et al., (PNAS, 2020), which studies related issues using computational modeling, information theory, and experiments. In that paper, the consequences of "distributed" kinetic proofreading, a model for the reset steps along the pathway, and experiments considering some spatial aspects of kinetic proofreading are considered.

4) I think it would be appropriate to cite the original kinetic proofreading papers by Hopfield and Ninio.

*Reviewer #2 (Recommendations for the authors):*

An estimate of the goodness-of-fit is important to provide some confidence that the model used to fit the experimental data is sound. Alternative models might be fit just as well and might provide very different conclusions otherwise; the data fits to KP 'by default'. It would also be helpful to be more explicit in the methods about how this fitting was done. Could the authors model the data assuming a series of reversible reactions (ie not KP), with the rate constant for each being sufficiently low to still fit the delay observed between receptor engagement and downstream output. How would this compare to the KP model?

---

## [Author Response]

Reviewer #1 (Recommendations for the authors):1) In the paragraph beginning with line 144: It should be noted that the kinetic proofreading model that the authors use is very simplified. The scaling with the power of steps, n, is only correct under certain approximations (see Hopfield's original PNAS paper). While the authors' note one of these approximations later (line 155), this is not the only one, and so mentioning the approximate nature of the analyses earlier on would be good.

We apologize for not clearly explaining our model’s limitations in the manuscript. We have incorporated the following paragraph on page 10 line 206 to better communicate the features of our simple kinetic proofreading model:

“Our model represents the expected output of a kinetic proofreading signaling network at steady-state. It assumes a network of sequential and non-reversible biochemical events where the rate of progression to the next step is much slower than the rate of reset upon ligand unbinding. We cannot determine the exact number of proofreading steps in the network, as the output of many weak proofreading steps could behave the same as the output of a few strong proofreading steps. However, if we assume equivalent strength for all proofreading steps, we can estimate a lower limit of the number of steps. Our model captures the expected output of a saturable signaling step downstream of such a proofreading system. When we apply our model to a measured signaling event in our system, we assume that event is downstream of the proofreading cascade and ask how many upstream steps of equivalent proofreading best explain its output.”

2) As Figure 3 shows, there is quite a bit of cell-to-cell variability. Can this variability data be quantified to provide insights into this "noise", which is important in determining the response of the population of cells?

Thank you for highlighting this feature of our dataset. We have included the following paragraph on page 7 line 153 discussing the sources of the cell-to-cell variability and how we leverage it to better explore the occupancy axis of our model.

“The lentiviral vectors we used to express our zdk-CAR and live-cell biosensors result in cell-to-cell variability in the expression level of each construct, even after sorting for cells with CAR expression levels similar to native TCR expression in T cells (Tischer and Weiner, 2019). To account for variability in biosensor expression, we normalize each cell to its baseline biosensor intensity measured in the absence of ligand engagement. The cell-to-cell variability in zdk-CAR expression has the benefit of expanding the receptor occupancies that we sample, as cells expressing more zdk-CAR receptors bind more LOV2 under all half-life conditions (Figure 3A, middle). If all cells expressed identical amounts of receptor, it would be laborious to generate and analyze the large number of independent CAR expressing clones needed to sample a wide range of receptor occupancies. Importantly, the variability in receptor expression is not unique to our CAR system; native T cells also vary in TCR expression level and can respond to antigen even at very low levels of TCR expression (Labrecque et al., 2001). Kinetic proofreading could enable cells to tolerate variability in receptor expression level, as it results in cells that are increasingly sensitive to ligand binding half-life instead of receptor occupancy.”

3) The authors may be interested in a paper by R. Ganti et al., (PNAS, 2020), which studies related issues using computational modeling, information theory, and experiments. In that paper, the consequences of "distributed" kinetic proofreading, a model for the reset steps along the pathway, and experiments considering some spatial aspects of kinetic proofreading are considered.

Thank you very much for calling this paper to our attention. In light of this suggestion, we have implemented the steady-state model from Ganti et al., to explore the plausibility of the number of proofreading steps our model fits to our datasets. From this work we have uploaded a new supplementary figure (Figure 4, figure supplement 4) with the figure caption on page 52. We discuss our exploration the model from Ganti beginning on page 14 line 314.

“To assess the plausibility of a larger number of proofreading steps, we implemented the steady state kinetic proofreading model from Ganti et al., (Ganti et al., 2020). The model estimates the minimum number of proofreading steps required to discriminate between cognate-ligands and self-ligands with different binding half-lives present at a given concentration ratios at a given Hopfield error-rate (Hopfield, 1974). First, we evaluated what combinations of ligand half-lives and concentration ratios an 11-step kinetic proofreading network could discriminate at an error rate less than 10^-3^ (Figure 4, figure supplement 4A). We chose the error rate of 10^-3^, as it is an order of magnitude less than the theorized 10^-4^ upper limit error rate of the native TCR (Ganti et al., 2020). At moderate half-life ratios, an 11-step network can discriminate cognate peptides present in small concentrations (e.g. 1 cognate-ligand per 1000 self-ligands at a half-life ratio of 6).

In our optogenetic system, the ratio of the average ligand binding half-life between the longest suppressive half-life and the shortest fully activated half-life is about 2. However, an 11-step network is insufficient to discriminate between ligands with a half-life ratio of 2, even at the high ligand ratio of 1 (equal concentrations of cognate- and self-ligand). This suggests our cells are unlikely to be detecting the average ligand binding half-life of each blue-light condition, but are more likely detecting longer-lived binding events from the underlying distribution of half-lives. Another possibility is that our in vitro washout measurements, which measure average ligand binding half-lives of soluble ligands diffusing in three dimensions, differ from the half-lives of ligand-receptor interactions between the cell’s plasma membrane and the supported lipid bilayer diffusing in two dimensions (J. Huang et al., 2010).

To better explore the kinetic proofreading model space, we generated heatmaps reporting the required number of steps to discriminate combinations of ligand and half-life ratios at an error rate of 10^-3^ (Figure 4, figure supplement 4B). To discriminate between ligands with a half-life ratio of two, at least 14 steps are needed when the ligands are at equal concentrations, and more than 25 steps are needed if cognate-ligands are 1 per 1000 self-ligands. The required number of proofreading steps decreases rapidly as the half-life ratio increases, reaching a minimum of 8-steps needed for a concentration ratio of 1/1000 and a half-life ratio of 10, which is more in line with physiological half-life ratios between agonist and non-agonist peptides (M. M. Davis et al., 1998).

After comparing our results with the Ganti model, this analysis suggest that our number of fit proofreading steps may be somewhat inflated as a function of our use the average ligand binding half-lives of three dimensional washout experiments in place of the two dimensional single molecule information T cells use to make activation decisions. However, the higher fit N values are more consistent with the required number of steps to discriminate ligands under more physiological conditions than our previous measurements of ~3 steps, which would not be expected to discriminate ligands with half-life ratio of 10 even at a ligand ratio of 1 (Figure 4, figure supplement 4B, right).”

4) I think it would be appropriate to cite the original kinetic proofreading papers by Hopfield and Ninio.

We apologize for omitting these key works. Citation of these papers existed in earlier versions, but we inadvertently deleted that section without bringing the references back elsewhere. This has been now corrected (page 2 line 43).

Reviewer #2 (Recommendations for the authors):An estimate of the goodness-of-fit is important to provide some confidence that the model used to fit the experimental data is sound. Alternative models might be fit just as well and might provide very different conclusions otherwise; the data fits to KP 'by default'.It would also be helpful to be more explicit in the methods about how this fitting was done.

We have expanded our methods section on model fitting on page 32 line 719 as reproduced below. We also moved this section to succeed relevant sections describing the data acquisition and processing methods for our occupancy and biosensor output measurements used before fitting the data.

“We fit each biological replicate dataset to the simple model of kinetic proofreading described previously (Figure 3D) (Tischer and Weiner, 2019). Each datapoint consists of an average ligand-binding half-life, a CAR occupancy measurement, and a biosensor output measurement (see above). With those three measured values, we fit the remaining three parameters of our model n, K, and β (number of proofreading steps, input for half-max biosensor output, and basal signaling output respectively) using the Levenberg–Marquardt least-squares algorithm as implemented by the curve_fit function from the python SciPy library (Virtanen et al., 2020).”

Could the authors model the data assuming a series of reversible reactions (ie not KP), with the rate constant for each being sufficiently low to still fit the delay observed between receptor engagement and downstream output. How would this compare to the KP model?

Importantly, our analysis is for the effects of different half-lives for signal outputs at stead-state and does not rely on the analysis of delays between steps. A series of reversible reactions would not be able to capture the dependance of signaling output we measured on ligand-binding half-life at steady state. Such a system’s output at steady state would depend on the steady-state receptor occupancy, with the reaction rate constants influencing how long it took the system to reach steady state, regardless of the number of reversible steps. We apologize for not emphasizing that our measurements are at steady state as opposed being based on temporal delays between steps following ligand binding. We thank the reviewers for suggesting that we compare with other models, as discussing the differences between kinetic proofreading systems and linear systems of reversible reactions has made our manuscript stronger. This discussion has been included on page 8 line 179, and reproduced below.

“To overcome this technical limitation of our system, we chose instead to measure the steady-state output of the antigen signaling cascade achieved several minutes after ligand binding. Kinetic proofreading systems behave differently than non-proofreading systems at steady-state. A non-proofreading system’s steady-state output is set by the number of ligand-bound receptors and not the binding half-lives of those ligands (Figure 3D, left). In contrast, a kinetic proofreading system can produce different steady-state outputs in response to ligands of different binding half-lives, even when ligand densities are adjusted to achieve equivalent occupancy (Daniels et al., 2006) (Figure 3D, right). Signaling events take varying amounts of time to occur after ligand binding (Lin et al., 2019; Yi et al., 2019). However, the temporal delays between steps are on the order of tens of seconds. By imaging the cells after minutes of constant exposure to a set ligand binding half-life, we measure the steady state output achieved at a signaling event in the cascade on a longer timescale than these delays (Tischer and Weiner, 2019).